# Neurite Outgrowth and Morphological Changes Induced by 8-trans Unsaturation of Sphingadienine in kCer Molecular Species

**DOI:** 10.3390/ijms20092116

**Published:** 2019-04-29

**Authors:** Seigo Usuki, Noriko Tamura, Tomohiro Tamura, Kunikazu Tanji, Daisuke Mikami, Katsuyuki Mukai, Yasuyuki Igarashi

**Affiliations:** 1Lipid Biofunction Section, Frontier Research Center for Advanced Material and Life Science, Faculty of Advanced Life Science, Hokkaido University, Sapporo 011-0021, Japan; dmikami@sci.hokudai.ac.jp (D.M.); yigarash@pharm.hokudai.ac.jp (Y.I.); 2National Institute of Advanced Industrial Science and Technology (AIST), Sapporo 061-8517, Japan; n-tamura@aist.go.jp (N.T.); t-tamura@aist.go.jp (T.T.); 3Department of Neuropathology, Institute of Brain Science, Hirosaki University Graduate School of Medicine, Hirosaki 036-8562, Japan; kunikazu@hirosaki-u.ac.jp; 4R&D Headquarters, Daicel Corporation, Tokyo 108-8230, Japan; kt_mukai@jp.daicel.com

**Keywords:** ceramide, konjac, semaphorin3A, neurite outgrowth, neuropilin1, endoglycoceramidase, sphingadienine

## Abstract

Konjac ceramide (kCer), which consists of plant-type molecular species of characteristic shingoid bases and fatty acids, is prepared from konjac glucosylceramide GlcCer by chemoenzymatical deglucosylation. kCer activates the semaphorin 3A (Sema3A) signaling pathway, inducing collapsin response mediator protein 2 (CRMP2) phosphorylation. This results in neurite outgrowth inhibition and morphological changes in remaining long neurites in PC12 cells. Whether a specific molecular species of kCer can bind to the Sema3A receptor (Neuropilin1, Nrp1) and activate the Sema3A signaling pathway remains unknown. Here, we prepared kCer molecular species using endoglycoceramidase I-mediated deglucosylation and examined neurite outgrowth and phosphorylation of collapsin response mediator protein 2 in nerve growth factor (NGF)-primed cells. The 8-trans unsaturation of sphingadienine of kCer was essential for Sema3A-like signaling pathway activation. Conversely, 8-cis unsaturation of kCer molecular species had no effect on Sema3A-like activation, and neurite outgrowth inhibition resulted in remaining short neurites. In addition, α-hydroxylation of fatty acids was not associated with the Sema3A-like activity of the kCer molecular species. These results suggest that 8-trans or 8-cis isomerization of sphingadienine determines the specific interactions at the ligand-binding site of Nrp1.

## 1. Introduction 

The population aged 60 and above is growing faster than all younger age groups around the world. The rapid growth of the aging population occurs not only in developed countries such as the United States (US), the European Union (EU), and Japan, but also in underdeveloped countries based on World Population Prospect estimates [1]. The rapidly aging population will increase medical expenses and place a considerable financial burden on the public universal health insurance system. The importance of dietary supplements for preventing the rise in healthcare costs is known, as they maintain and promote healthy life years and reduce the risk of various lifestyle-associated diseases. To improve the healthy life expectancy of the population, various dietary ingredients derived from natural sources have been investigated and used as dietary supplements for disease prevention. 

Ceramides have been studied as supplements to prevent lifestyle diseases [2]. Ceramides are composed of sphingosine and fatty acids, and frequently contain sphingolipids such as glycosphingolipids, sphingomyelins, and free ceramides on the mammalian cell membrane [3]. These molecules can act as signaling molecules to regulate cellular functions including cancer growth, diabetes, and cardiovascular diseases [4,5,6]. In plants, free ceramides, which we refer to as ceramides herein, occur in lower abundance than other sphingolipids such as sphingomyelin and glucosylceramide (GlcCer), which is enriched in plants [7]. Food sphingolipids such as GlcCer, which is found in cereals [7], may have beneficial effects by preventing metabolic syndrome and related diseases [8,9]. Konjac (*Amorphophallus konjac*, K. Koch) is a food plant that is rich in GlcCer. The efficacy of konjac GlcCer (kGlcCer) for improving transepidermal water loss in mice and humans has been studied by Uchiyama et al. [10], and kGlcCer is used as a health food and in cosmetics. However, it remains to be elucidated whether kGlcCer and its metabolites have a direct effect on itching hypersensitivity of the skin caused by extra-neurite invasion into the stratum corneum [11], which is often seen in itch-causing skin diseases such as atopic eczema and *Psoriasis vulgaris* [12]. Recently, we chemoenzymatically synthesized konjac ceramide (kCer) by deglucosylation of kGlcCer using endoglycoceramidase I (EGCase I), which cleaves the β-glycosidic bond in GlcCer [13]. Unlike kGlcCer and other animal-type ceramides, exogenous kCer has a neurite outgrowth inhibitory effect. Neurite outgrowth is a crucial morphological event in neuronal differentiation [14], which begins at the cell body and extends outward to form a functional synapse [15]. Elongated neurites are directed to their targets by sensing attractive and repulsive molecules through receptors expressed on the growth cone [16]. In our previous study [17], we showed that, in PC12 cells, kCer shows similar changes in cell morphology to a spindle shape and short neurite retraction as semaphorin 3A (Sema3A). Nerve growth factor (NGF)-related trkA activity is associated with neuronal differentiation and neurite outgrowth [18], and the trkA signaling pathway is activated by rapid tyrosine phosphorylation of trkA and the consequent activation of extracellular signal-regulated kinases (ERKs)/mitogen-activated protein kinases (MAPKs) [19]. However, the effect of kCer is not associated with activation of the trkA signaling pathway, but rather activation of the Sema3A signaling pathway [20]. Sema3A acts as a repulsive factor for neuronal outgrowth in peripheral neurons and has the opposite function to NGF in the stratum corneum of human heathy skin. 

We were interested in examining the effect of kCer on sensory nerves and keratinocytes because they also express the Sema3A receptor neuropilin1 (Nrp1). kCer binds to Nrp1 in neurons and keratinocytes in PC12 cells [21] and HaCaT cells [20]. Sema3A suppresses the migration of keratinocytes toward serum-derived chemoattractants [22]. We demonstrated that the effect of kCer on keratinocyte migration is associated with activation of the Sema3A signaling pathway [20]. Histamine (His) stress promotes immature keratinocyte migration [23] and plays an important role in inflammation and nervous irritability. It also regulates the expression of pruritic factors such as NGF and Sema3A in skin keratinocytes via the His1 receptor (H1R) [24,25]. We demonstrated that kCer does not interact with the His receptors H1R or H4R via activation of G-protein-coupled receptors on the cell surface [20].

Although animal-type ceramides containing sphingosine (d18:1) and dihydrosphingosine (d18:0) have been studied extensively [26], it remains unclear whether plant-type ceramides containing sphingoid bases such as 4, 8-sphingadienine (d18:2) and 4-hydroxy-8-sphingenine (t18:1) have biological activity (Figure 1A) [27,28]. Also, it is unknown whether kCer molecular species contain active or inactive ones that specifically contribute or do not contribute to kCer-induced Sema3A activity. In the present study, we investigated the Sema3A-like neurite outgrowth retraction activity in the presence or absence of different kCer molecular species. 

## 2. Results

### 2.1. EGCase I Treatment

Each of the kCer molecular species was enzymatically synthesized from kGlcCer. Despite the insolubility of GlcCer, the efficiency of its enzymatic hydrolysis was improved by up 50% by: (1) using highly concentrated EGCase produced from an expression vector in actinomycetes that allows efficient EGCase I expression and export into the culture medium without protein aggregation, which is achieved by delaying protein synthesis and allowing secretion to proceed at low temperature as previously reported [17]; and (2), by performing a second EGCase I reaction after drying the lower layer of the Bligh–Dyer extract of the first enzymatic reaction mixture without detergent, which enabled the almost complete conversion of the remaining kGlcCer to kCer. Each kCer molecular species (d18:2^4t,8c^-C16h:0, d18:2^4t,8t^-C16h:0, d18:2^4t,8c^-C18h:0, d18:2^4t,8t^-C18h:0, d18:2^4t,8c^-C20h:0, d18:2^4t,8t^-C20h:0, t18:1^8c^-C22h:0, t18:1^8c^-C23h:0, and t18:1^8c^-C24h:0) was prepared in this way (Figure 1B and Figure 2B). One unit of EGCase I was defined as the amount of enzyme that hydrolyzes 1 µmol of kGlcCer per min as a donor substrate. Under these conditions, the enzyme unit per tube used was 10.7 mU (kGlcCer, 90 nmol; 250 µg of enzyme protein; 16 h incubation). The different kGlcCer molecular species showed no differences in EGCase I reactivity or yield.

### 2.2. Preparation of d18:2^4t,8t^-C16:0 and d18:2^4t,8c^-C16:0

Plant type-ceramides are mostly composed of α-hydroxyl fatty acid. To test whether the presence of α-hydroxylation of fatty acid is essential for the activity, we had to prepare plant-type sphingoid bases from plant GlcCer, followed by synthesizing non-hydroxy plant ceramides using non-hydroxylated fatty acid. Sphingoid bases were prepared from kGlcCer, and 8-trans and -cis isomers of sphingadienine were separated by octadesyl silyl column-equipped reverse phase high performance liquid chromatography (ODS-HPLC) (Appendix A). Sphingadienines d18:2^4t,8c^ and d18:2^4t,8t^ had purities of 98.0% and 95.0%, respectively (Appendix A). N-palmitoylation was performed for each sphingadienine, followed by silica gel column purification, which resulted in a purity of 90% for the N-palmitoylated products (Appendix A). We detected a single band on the TLC plate (Appendix A), and confirmed *N*-palmitoylation of sphingadienine by infusion ESI-MS (Appendix A). Synthesized ceramides had *m*/*z* values of 536.5 (precursor ion; Appendix A), and 262.3 (product ion; Appendix A). Finally, we obtained *N*-palmitoyl-d18:2^4t8c^ (2.0 mg, 3.7 µmol) and *N*-palmitoyl-d18:2^4t8t^ (1.0 mg, 1.9 µmol). 

### 2.3. Effect of kCer Molecular Species on Neurite Inhibitory Activity and Cell Shape 

The nine kCer molecular species (Figure 2B) showed no cytotoxicity up to 50 µM cytotoxicity, similar to kCer as previously reported [17]. 

PC12 cells were induced to differentiate by the addition of NGF in the presence of kCer or Sema3A, and cultures were fixed and stained with Coomassie Brilliant Blue (CBB) as described in Materials and Methods (Figure 3). Single treatment with kCer or Sema3A did not induce neurite outgrowth or morphological changes in cells (Figure 3A, second and third images in second row from the top). kCer or Sema3A treatment inhibited neurite extension in NGF-primed cells, resulting in a few remaining long neurites and loss of most short neurites (Figure 3A, second and third images in the top row). In addition, kCer treatment caused a change in cell morphology to a spindle shape compared with the control culture; NGF treatment alone induced a star-shaped morphology. The morphological changes induced by kCer and Sema3A were indicated by the remaining long neurites in distribution plots (Figure 3B, second from the top row). Cells treated with three kCer molecular species showed that trans unsaturation at C8 of sphingadienine (d18:2 4t,8t) with a fatty acid moiety (C16h:0, C18h:0, and C20h:0) was associated with remaining Sema3A-like long neurites and changes in cell morphology to a spindle shape (Figure 4A,B). Conversely, their cis counterpart (d18:2 4t,8c) showed long and short neurite outgrowth inhibitory activity (Figure 4B, second row from the top), although the inhibition was not accompanied by changes in morphology to a spindle-like shape or loss of short neurites (Figure 4A,B). Trans monounsaturation at C8 of phytosphingenine (t18:1 8c) with a fatty acid moiety (C22h:0 and C24h:0) showed no inhibitory activity (Figure 4B, second and third graphs in the top row) and did not induce morphological changes in NGF-primed cells (Figure 4A, two images in the right column). The phytosphingenine with C23h:0 did not show neurite outgrowth inhibitory activity. 

To determine whether α-hydroxylation of the fatty acid moiety was necessary for activity, the cis/trans unsaturation isomer with a C16:0 fatty acid was compared by assessing neurite outgrowth inhibition. As shown in Figure 5, α-hydroxylation at the C16 fatty acid was not necessary for the morphological changes of remaining long neurites and neurite inhibitory activity. In addition, monounsaturation at C4 of sphingenine (d18:1 4t), which is a major sphingoid base observed in mammals, had no effect on neurite inhibitory activity or cell morphological changes (Figure 5B, left). 

### 2.4. Collapsin Response Mediator Protein 2 (CRMP2) Phosphorylation

Phosphorylation of CRMP2 is essential for Sema3A activity. To examine the effects of the kCer molecular species on CRMP2 phosphorylation, NGF-primed cells were treated with each species (d18:1^4t^-C16:0, d18:1^4t^-C16h:0, d18:2^4t,8c^-C16:0, d18:2^4t,8c^-C16h:0, d18:2^4t,8t^-C16:0, and d18:2^4t,8t^-C16h:0). Immunoblot analysis showed that trans unsaturation of sphingadienine (d18:2 4t,8t) was associated with CRMP2 phosphorylation activity independent of the presence or absence of α-hydroxylation of the C16 fatty acid (Figure 6A, lane 5 and 6). However, species with monounsaturation of sphingenine (d18:1 4t) and cis unsaturation of sphingadienine (d18:2 4t,8c) showed no CRMP2 phosphorylation activity, and α-hydroxylation of the C16 fatty acid did not contribute to the activation.

### 2.5. Sema3A Receptor-Binding Activity

To determine whether kCer interacts with Sema3A for binding to a cell surface receptor on PC12 cells, the cells were incubated with alkaline phosphatase-fused Sema3A (AP-Sema3A) in combination with unlabeled Sema3A or kCer. As shown in Figure 6B, the binding of AP-Sema3A to cells was partially inhibited by the addition of 10–50 µM of a kCer species with trans unsaturation of sphingadienine (d18:2 4t,8t). Conversely, species with cis unsaturation of sphingadienine (d18:2 4t,8t) did not inhibit binding.

*Nrp1* gene silencing in PC12 cells resulted in the disappearance of AP-Sema3A (Figure 6C), whereas trans or cis unsaturation of sphingadienine had no AP-Sema3A-binding activity.

## 3. Discussion

Plant-type ceramides differ from animal-type ceramides in the composition of sphingoid bases, in particular the position of double bonds and the α-hydroxyl group of fatty acids (Figure 1). Nine plant-type sphingoid bases (Figure 1A) have been identified as major sphingoid bases of plant sphingolipids. Free ceramides, such as plant-type ceramides, are minor components of plant sphingolipids compared with the composition of animal-type ceramides. Plant-type ceramides are derived from GlcCer or glycosylinositol phosphoceramide (GIPC), which have a different composition of sphingoid bases. A high unsaturation rate (≥50%) of sphingoid bases in GlcCer is due to the existence of d18:2 (sphinganine with diunsaturation at 4t, 8t or 4t, 8c) and t18:1 (phytosphingosine with monounsaturation at 8t). On the other hand, GIPC is composed of up to 90% sphingoid bases such as phytosphingosine with saturation (t18:0), and a lower amount of d18:0 bases. Thus, plant-type GlcCer and GIPC possess distinct compositions of sphingoid bases, particularly regarding the degree of unsaturation. In addition, the cis/trans unsaturation ratio at the C8 carbon differs between GlcCer and GIPC: the cis/trans ratio is 4.0 in GlcCer and 0.5 in GIPC. These characteristics of plant sphingoid base unsaturation in different glycosphingolipid classes are consistent with results previously reported by Ohnishi [29], and indicate that GlcCer is enriched in cis unsaturated sphingoid bases, whereas GIPC is enriched in saturated or trans unsaturated sphingoid bases. The ceramide fraction of kCer is abundant in the saturated sphingoid base of t18:0, similar to GIPC, although it also contains d18:2 and t18:1 species to some extent, with a high cis/trans ratio close to that of GlcCer. This indicates that the plant-type ceramide molecules of kCer are direct substrates of GlcCer and GIPCs, or that they are deglycosylated products of the complex sphingolipids (Figure 1B). In our experiment, we prepared kCer molecular species by deglucosylation of kGlcCer using EGCase I (Figure 2A). The free ceramide composition of konjac extracts differs from that of kCer, as kCer is produced from kGlcCer, and its sphingoid base content is therefore similar to that of kGlcCer (Figure 2B). In this study, kCer molecular species in the konjac extract were plant-type ceramides; therefore, activity analysis using kCer molecular species indicated the potential activity of konjac extracts.

Binding of guidance molecules to receptors on the growth cone of neurites leads to the activation of intracellular signaling cascades [30,31]. In turn, this receptor-binding process gives rise to dynamic changes in the cytoskeleton [32] and subsequent directional neurite outgrowth and target recognition [33]. Sema3A is a class 3 secreted member of the semaphorin family [34] that acts as a neurite-repulsion factor by competing with NGF for the regulation of the process via which axonal growth cones are directed and extended to their proper targets. 

In the nerve endings of peripheral neurons in the human skin, Sema3A acts as a repulsive factor against excess neuronal outgrowth and has the opposite function to NGF in the stratum corneum [35]. The balance, or ratio, of activity of extracellular levels of Sema3A and NGF regulates skin barrier maintenance, preventing sensory nerves from extending neurites, which results in skin hypersensitivity [36]. This balance of Sema3A and NGF is maintained by secretory Sema3A and the NGF of keratinocytes. Extracellular Sema3A binds to Nrp1 expressed in nerve cells and keratinocytes, and the nerve endings of peripheral neurons, activating the Sema3A signaling pathway. In keratinocytes, Sema3A signaling activation suppresses cell migration and cell differentiation. Sema3A signaling activation in nerve cells causes neurite outgrowth retraction. We previously reported this finding using HaCaT cells [20]. In addition, we found increased phosphorylation of CRMP2 and microtubule depolymerization in PC12 cells [37], as well as increased phosphorylation of Cofilin. 

Here, the kCer-mediated inhibition of neurite outgrowth and cell migration was shown in PC12 and HaCaT cells. kCer binds to Nrp1 as a Sema3A-like agonist, resulting in Nrp1/PlexA complex formation and activation of Sema3A signaling. In addition, kCer and Sema3A inhibited His-enhanced migration of immature HaCaT cells [20]. We showed that kCer does not interact with the histamine receptor H1R or H4R directly; however, we speculate that kCer may transduce a signal downstream of the His signaling pathway. 

As previously reported by us [21], kCer had higher binding affinity for Nrp1 (K_D_ 5.057 μM) than those of animal ceramides such as C16Cer, C18Cer, and C24Cer, with *k*_a_ values (M^−1^·s^−1^) > 1 × 10^5^ and high *k*_d_ values. kCer molecular species differ from animal ceramides in the cis/trans unsaturation at C8 (d18:2 at 4t, 8t or 4t, 8c, sphingadienine) and cis monounsaturation at C8 (t18:1, phytosphingenine). In this study, the activity of kCer molecular species was compared with that of Sema3A by examining the specific neurite length distribution of NGF-primed PC12 cells. The Sema3A-like activity of kCer was characterized by the morphological changes of PC12 cells to a spindle-like shape. As shown in Figure 4, we demonstrated that trans unsaturation at C8 of sphingadienine is essential for the spindle-like shape of PC12 cells with the acyl chains C16h:0, C18h:0, and C20h:0. Conversely, cis unsaturation at C8 of sphingadienine caused a neurite inhibitory effect on NGF-primed cells, whereas analysis of cell morphology showed round cells with short neurites. The inhibition of NGF-primed neurite outgrowth by C8 cis sphingadienine may be influenced by the cell toxicity of this ceramide species. At more than 50 μM, ceramides showed some cell toxicity by perturbating the cell membrane. This influence is often induced in a lipid structure-peculiar, non-specific manner due to the insolubility. 

On the other hand, trans monounsaturation at C8 of phytosphingenine had no neurite inhibitory effect and did not change the morphology of cells with the acyl chains C22h:0 and C24h:0. 

In addition to the sphingoid bases of kCer molecular species, another plant-specific feature of sphingolipids is that α-hydroxyl fatty acids are mostly restricted to acyl chain lengths of C16h:0, C18h:0, C20h:0, and C22h:0 in kCer molecular species. The Sema3A-like activity of kCer molecular species is due to the α-hydroxyl fatty acid and sphingoid bases with trans unsaturation at C8 of sphingadienine. To determine whether the presence of the α-hydroxyl fatty acid moiety is required for the Sema3A-like activity of kCer molecular species, the fatty acid moiety in kCer molecular species was substituted with the C16:0 fatty acid. The results showed that α-hydroxylation of fatty acids is not necessary for the Sema3A-like activity of kCer molecular species. 

The specificity of the Sema3A-like activity of kCer was examined by assessing CRMP2 phosphorylation of NGF-primed cells during Sema3A signaling pathway activation triggered by binding of kCer to Nrp1 on the PC12 cellular membrane. In the same C16 acyl chain, modification of the α-hydroxylation of the fatty acid and cis/trans unsaturation at C8 of sphingadienine showed that α-hydroxylation of fatty acids is not essential, whereas trans isomer unsaturation of sphingadienine is essential for CRMP2 phosphorylation (Figure 6A). Cis isomer unsaturation did not show CRMP2 phosphorylation activity.

kCer binding to cell surface receptors was shown by the binding displacement profile of FITC-Sema3A and AP-Sema3A by kCer [20,21]. In this study, trans unsaturation of sphingadienine inhibited AP-Sema3A binding at 10–50 µM, (Figure 6B) whereas cis unsaturation of sphingadienine did not. In addition, silencing of the *Nrp1* gene in PC12 cells prevented AP-Sema3A binding (Figure 6C). On the other hand, neither cis nor trans unsaturation of sphingadienine affected the lack of AP-Sema3A binding at 50 µM. Regardless of the presence or absence of α-hydroxylation of the C16 fatty acid, animal-type C16Cer did not show Sema3A-like activity for NGF-primed PC12 cells. Table 1 shows a summary of the structure–activity relationship of kCer molecular species. C16 and C18 fatty acid commonly carbon chain length enriched in plant ceramides. 

Oral intake of dietary GlcCer improves the skin barrier function by preventing transepidermal water loss in mice and healthy human subjects [10]. GlcCer is metabolized to ceramides and sphingoid bases in the digestive tract, followed by intestinal mucosal absorption and localization to lymphatic ducts [38]. The accumulation of sphingoid bases in lymphatic ducts is controlled by sphingoid base uptake into intestinal cells and metabolism. Further metabolism of dietary sphingolipid bases by ceramide synthase and GlcCer synthase or sphingomyelinase, yields sphingolipids that can localized to the skin, where kCer molecular species might appear as ceramides with non-hydroxyl fatty acid. In the present study, we speculated that oral intake of kCer would result in the appearance of kCer as molecules with 4t, 8t sphingadienine and non-hydroxyl fatty acids possessing Nrp1-binding activity and Sema3A signaling pathway activation capacity.

## 4. Materials and Methods

### 4.1. General

The following materials were commercially obtained: the kGlcCer and kGlcCer molecular species d18:2^4t,8c^-C16h:0, d18:2^4t,8t^-C16h:0, d18:2^4t,8c^-C18h:0, d18:2^4t,8t^-C18h:0, d18:2^4t,8c^-C20h:0, d18:2^4t,8t^-C20h:0, t18:1^8c^-C22h:0, t18:1^8c^-C23h:0, and t18:1^8c^-C24h:0 were purchased from Nagara Science Co., Ltd. (Gifu, Japan); 2.5S mouse NGF (N100NF4325) was from Alomone Labs, Jerusalem, Israel; semaphorin 3A (Sema3A, 193-17051) was from Wako Corp. (Tokyo, Japan); anti-CRMP2 pAb and anti-phospho-CRMP2 pAb (pthr509) were from Sigma-Aldrich (St. Louis, MO, USA). Alkaline phosphatase (AP) activity was determined using an ALP activity assay kit (LabAssay™, Takara, Shiga, Japan) with *p*-nitrophenylphosphate (pNPP, Sigma) as a substrate.

kCer was prepared in our laboratory based on a published procedure [17]. kCer is a plant-type ceramide molecular species containing sphingoid bases such as 4-trans-8-cis-sphingadienine (d18:2^4t,8c^), 4-trans-8-trans-sphingadienine (d18:2^4t,8t^), 4-hydroxy-8-cis-sphingenine (t18:1^8c^), and 8-cis-sphingenine (d18:1^8c^). The tested lipid kCer (or other ceramides: C2Cer, C16Cer, C18Cer) was dissolved in 0.025% bovine serum albumin (BSA)/Dulbecco’s modified Eagle’s medium (DMEM). All experiments were performed with approval from the regulatory boards of Hokkaido University.

### 4.2. Neurite Outgrowth Assay

PC12 cells (CVCL_J438) were grown under 5% CO_2_ at 37 °C in DMEM supplemented with 1% penicillin/streptomycin, 10% horse serum (HS), and 10% fetal bovine serum (FBS) in culture dishes. For neurite outgrowth activity, PC12 cells were stimulated with 100 ng/mL 2.5S NGF in serum-free medium in 0.01% poly-l-lysine-coated 12-well plates and incubated for up to 3 days. Subsequently, cell-cultured wells were gently overlaid by 2% glutaraldehyde in phosphate-buffered saline solution, followed by fixing cells at room temperature for 20 min. The glutaraldehyde solution was changed to 1% Coomassie Brilliant Blue G-250 (CBB) solution (1% CBB in 50% methanol/PBS), followed by staining for 2 h at room temperature. The cells were destained by 50% methanol/PBS and water, respectively. Bright-field cell images were obtained using color phases-based images from BXZ-700 microscopy (KEYENCE, Osaka, Japan) at 20-fold magnification. The length of each neurite was measured using imaging software (ImageJ 1.50g) and classified over a range of 10–150 μm, followed by conversion into cumulative ratios; y-axis values, mean % of neurites with length over the x-axis values previously as described [37]. All counting of the number of neurites with lengths of 10 to 150 μm is shown as 100% and calculated for each of x-axis valued-neurite length.

Before the neurite outgrowth experiment, NGF was dissolved in 0.025% BSA/DMEM containing each molecular species of kCer (50 µM).

### 4.3. CRMP2 Phosphorylation Analysis by Western Blotting

PC12 cell monolayers in 6-well plates were treated with NGF (100 ng/mL) for 48 h, then replaced with fresh DMEM containing either molecular species of kCer (50 µM) or Sema3A (250 ng/mL), and incubated for 24 h. Subsequent to the incubation, cells were used for western blot. Briefly, cells were lysed by RIPA buffer (Wako Corp.) containing supplements: cOmplete proteinase inhibitor (Roche, Basel, Switzerland) and PhosSTOP phosphatase inhibitor cocktail (Roche). The protein concentrations of the lysates were determined using the bicinchoninate protein assay kit (Nacalai Tesque, Kyoto, Japan). Equal amounts of protein (10 μg) were applied on sodium dodecyl sulfate polyacrylamide gel electrophoresis (SDS-PAGE, SuperSep™Ace, 5–20%, Wako Crop.), separated, and transferred to polyvinylidene fluoride (PVDF) membranes (Millipore Corp, Bedford, MA, USA) using a Trans-Blot SD Semi-Dry Electrophoretic Transfer Cell (BioRad, Berkeley, CA, USA). After blocking for 1 h with Blocking One (Nacalai Tesque), the membranes were incubated overnight at 4 °C with the following primary antibodies: anti-CRMP (1:2000), anti-pCRMP (1:1000), or anti-GAPDH (1:3000) in 10% Blocking One solution with 0.05% Tween 20 and 50 mM Tris-buffered saline (TBST). The next day, each membrane was washed with TBST three times for 10 min each and incubated with the corresponding horseradish peroxidase-labeled secondary antibodies (1:2000 to 1:4000) for 1 h at room temperature. The membrane was then treated for 1 min with chemiluminescent reagent (Chemi-Lumi One Super, Nacalai Tesque). We detected the antibody-bound protein bands using the Syngene G:BOX Chemi XT4 fluorescence and chemiluminescence gel imaging system (Syngene, Bangalore, India). The relative intensities of developed bands were quantified using JustTLC software (SWEDAY, Sodra, Sweden) and normalized to the loading control (GAPDH).

### 4.4. Receptor-Binding Study Using AP-Sema3A

A ligand-cell surface receptor binding assay was performed using alkaline phosphatase-fused Sema3A (AP-Sema3A). AP-Sema3A plasmid (chick origin) was obtained from Addgene (Watertown, MA, USA, plasmid # 29448). Recombinant AP-Sema3A protein was expressed in HEK293 cells that were transiently transfected with an expression plasmid using a Lipofection Kit (ScreenFecT™A plus, Wako Corp.). After 2 days of culturing in Opti-MEM, AP-Sema3A protein was concentrated in the conditioned medium up to a hundredfold, and the enzyme-specific activity was decided using an AP Activity Kit (LabAssay™ALP, Wako Corp.) and a SEMA3A ELISA Kit (ABIN1566604, Aviva Systems Biology, San Diego, CA, USA). According to the results of both assays, the ratio of relative AP activity units (APU, mmol/min/mg of protein) of AP per Sema3A protein was calculated as 15.0 APU. For the PC12 cell surface receptor-binding assay, AP-Sema3A was used at a final concentration of 25 nM in 6 × 10^4^ cells per well of a 24-well microplate. The 24-well plate was incubated at 22° C for 90 min, and the wells were washed with HH buffer (0.005% BSA/Hank’s balanced buffer + 20 mM HEPES buffer (pH 7.0)), followed by fixing with 3.7% formaldehyde/PBS solution. The fixation medium was exchanged with AP buffer (100 mM NaCl + 5 mM MgCl_2_/50 mM Tris-HCl (pH 9.5)). PC12 cell-derived AP activity was inactivated by incubating at 65 °C for 30 min. The wells were washed with HH buffer and treated with a substrate solution of BCIP/NBT Phosphatase Substrate (1-Component, KPL, Gaithersburg, MD, USA). After the coloration reaction with AP activity, the dye corresponding to AP-Sema3A was measured using ImageJ (1.50 g). The ligand substitution of cell surface receptor experiment was performed using 25 nM AP-Sema3A together with kCer and other ceramides dissolved in 0.3% BSA. 

### 4.5. RNA Interference and Transfection

Gene silencing was performed using siRNA. *Nrp1*-specific siRNA was purchased from Invitrogen (RSS332427) with the sense strand sequence of 5′-GCACCUACAUCAUCUUUGCACCAAA-3′. To compare the efficiency of *Nrp1* knockdown, scrambled siRNA (medium GC Duplex, Invitrogen) was used as a negative control. Transfection of siRNA was carried out using Lipofectamine™ 2000 (Invitrogen, Grand Island, NY, USA) according to the manufacturer’s instructions. Briefly, siRNA and Lipofectamine™ 2000 reagent were mixed in Opti-MEM (Invitrogen) and incubated for 5 min at room temperature to allow complex formation. Cells were washed with Opti-MEM, and the transfection mixture was added. The cells were incubated for 6 h after transfection, washed, and cultured for 24 h in complete medium containing 10% FCS and 10% HS. The silencing efficacy was evaluated by PCR and western blotting as previously described [21]. 

### 4.6. Ceramide Synthesis and LC-ESI-MS/MS Analysis

Sphingoid bases were prepared from konjac glucosylceramide as described previously [39]. 4-trans, 8-cis sphingadienine (d18:2^4t,8c^) and 4 trans, 8 trans-sphingadienine (d18:2^4t8t^) were isolated by ODS-HPLC as described previously [40]. To condense sphingoid base and palmitic acid, d18:2^4t8c^ (3.3 mg, 11.1 µmol), palmitic anhydride (7.2 mg, 11.1 µmol), and methanol (2 mL) were transferred to screw-capped glass vials and incubated at 37 °C for 22 h. Similar to the procedure described above, d18:2^4t8t^ (1.3 mg, 4.4 µmol), palmitic anhydride (2.8 mg, 5.7 µmol), and methanol (2 mL) were transferred to screw-capped glass vials and incubated at 37 °C for 22 h. The resulting ceramides were purified with an ODS-4 column (particle size, 5 µm; diameter, 10 mm; length, 150 mm; GL Science, Tokyo, Japan). Methanol (100%) was used as the mobile phase, and compounds were eluted at 3 mL/min. Eluents were monitored at ultraviolet absorbance of 210 nm. Fractionated materials were dissolved in methanol and analyzed by infusion electrospray ionization-tandem mass spectrometry (ESI-MS/MS) analysis using the TripleTOF 5600 system (AB SCIEX, Foster City, CA, USA).

### 4.7. Statistical Analysis

The number (*n*) in each experimental condition is shown in the figure legends. Data were analyzed statistically using the commercial program Prism 4.0 (GraphPad, San Diego, CA, USA). In comparisons of two experimental conditions, statistical analysis was performed using a paired *t* test. Alternatively, statistical analysis was done using one-way ANOVA followed by Tukey’s multiple comparison post-test and Dunnett’s test. A *p* value of < 0.05 was considered significant. * indicates significantly different results. Ranges of *p* values are indicated in the figure legends. NS indicates non-significant.

## Figures and Tables

**Figure 1 ijms-20-02116-f001:**
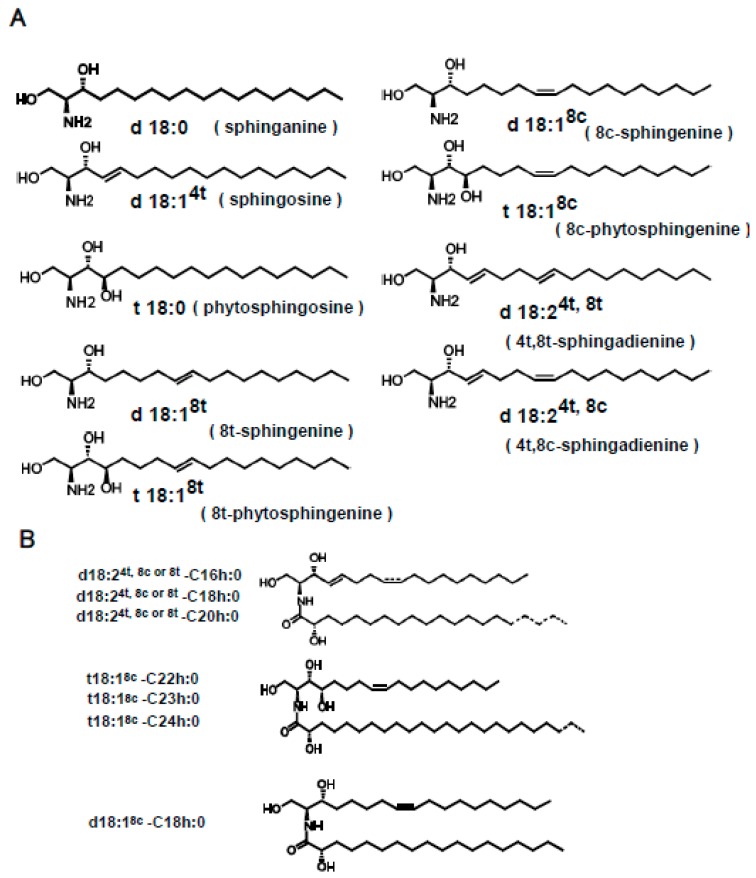
The nomenclature of sphingoid bases and ceramides followed the recommendations of the IUPAC-IUBMB Joint Commission. (**A**) Main sphingoid bases found in plants. (**B**) Molecular species of konjac ceramide (kCer) that are produced by endoglycoceramidase (EGCase I) treatment of konjac GlcCer (kGlcCer). Either cis or trans isomerism of sphingoid bases is shown in the broken line. The carbon chain length of the hydroxyl fatty acids C16 to C20 and C22 to C24 is also shown in the broken line.

**Figure 2 ijms-20-02116-f002:**
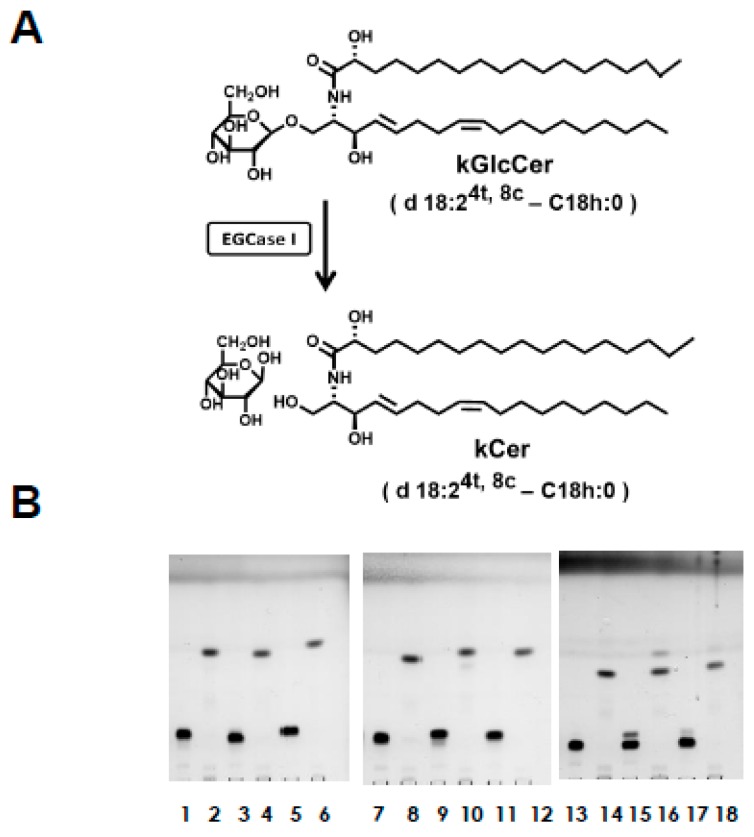
(**A**) EGCase reaction of kGlcCer. Plant-type ceramides can be prepared from plant-type GlcCer by EGCase I treatment. (**B**) TLC plate of Bligh–Dyer extracts before and after EGCase I reaction using kGlcCer molecular species, as indicated by the following lane numbers: 1 and 2, d18:2^4t,8c^-C16h:0; 3 and 4, d18:2^4t,8t^-C16h:0; 5 and 6, d18:2^4t,8c^-C18h:0; 7 and 8, d18:2^4t,8t^-C18h:0; 9 and 10, d18:2^4t,8c^-C20h:0; 11 and 12, d18:2^4t,8t^-C20h:0; 13 and 14, t18:1^8c^-C22h:0; 15 and 16, t18:1^8c^-C23h:0; 17 and 18, t18:1^8c^-C24h:0. The enzyme reaction was performed at 37 °C in 0.1 M sodium acetate buffer at pH 5.0. After the enzyme reaction, one additional EGCase I treatment was performed for the dried sample of Bligh–Dyer extracts without any detergent. The secondary dried samples were examined by TLC using chloroform:methanol:acetic acid (65:10:0.3, *v*/*v*) as the solvent. TLC plates were sprayed with 10% cupric sulfate in 8% phosphoric acid, and heated at 180 °C.

**Figure 3 ijms-20-02116-f003:**
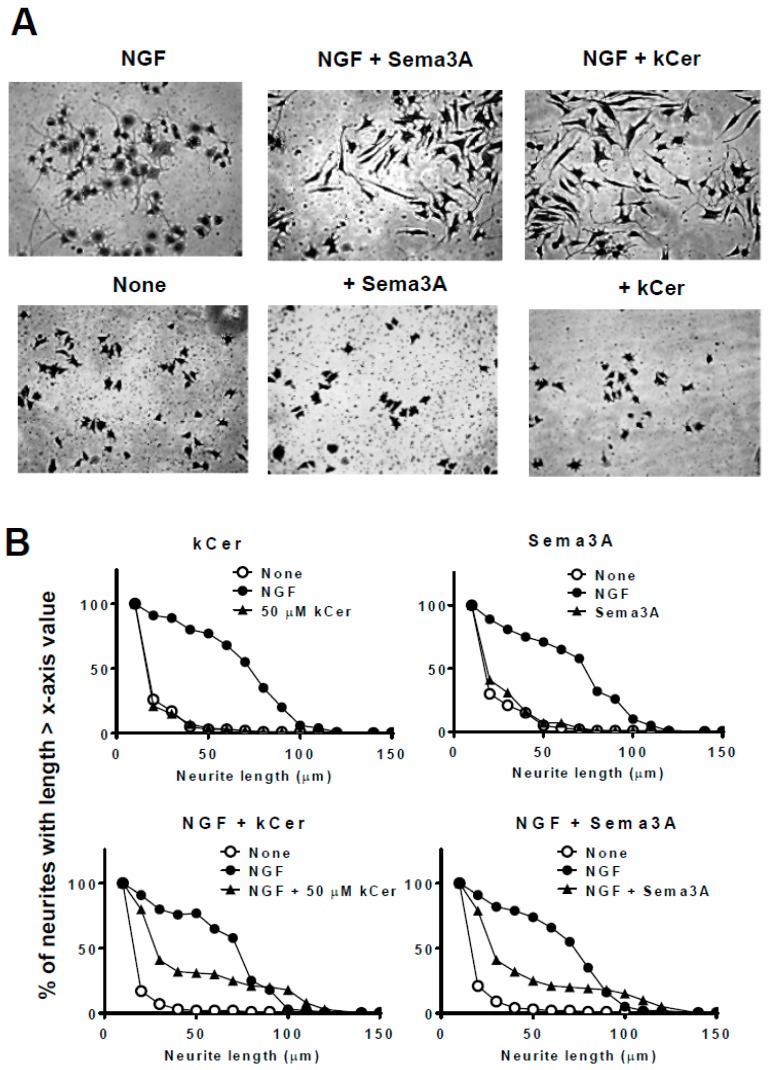
Effect of kCer and semaphorin 3A (Sema3A) on neurite outgrowth, neurite length distribution, and cell morphological changes in nerve growth factor (NGF)-primed PC12 cells. (**A**) Representative images of NGF (100 ng/mL, 2.8 nM in medium)-primed PC12 cells (7.5 × 10^4^ per mL) incubated in the absence or presence of NGF and 50 µM kCer or Sema3A. Cells were stained by 1% CBB and photographed (20× magnification). Scale bar: 100 μm. (**B**) Neurite length distribution changes of PC12 cells incubated with 50 µM kCer or 2 nM Sema3A. All counting of the number of neurites with lengths of 10 to 150 μm is shown as 100% and calculated for each of x-axis valued neurite length.

**Figure 4 ijms-20-02116-f004:**
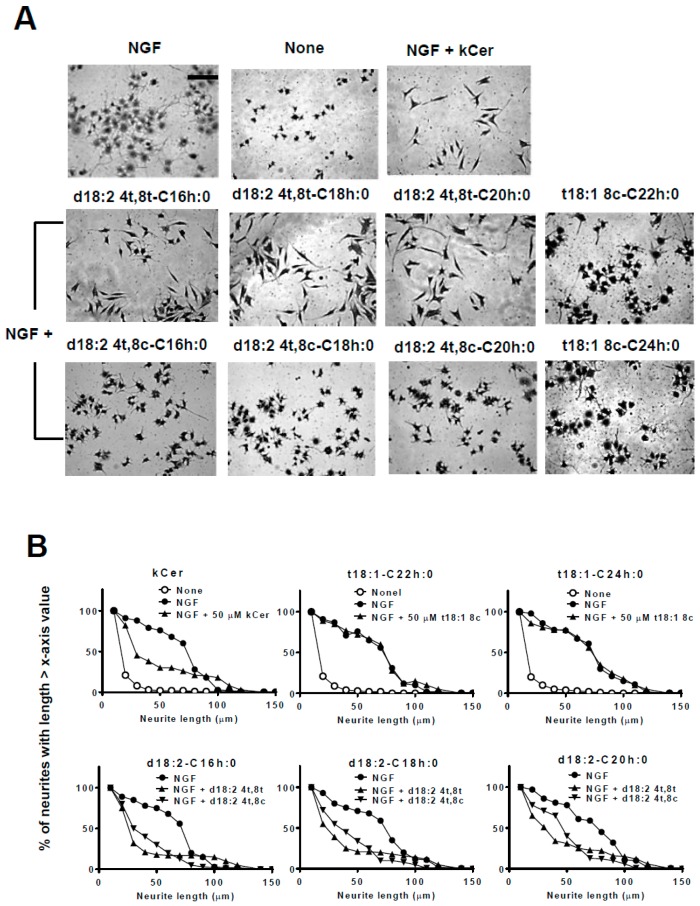
Effect of kCer molecular species on neurite outgrowth, neurite length distribution, and cell morphological changes in NGF-primed PC12 cells. (**A**) Representative images of NGF (100 ng/mL, 2.8 nM in medium)-primed PC12 cells (7.5 × 10^4^ per mL) incubated in the absence or presence of NGF and 50 µM kCer or kCer molecular species (d18:2^4t,8t^-C16h:0, d18:2^4t,8c^-C16h:0, d18:2^4t,8t^-C18h:0, d18:2^4t,8c^-C18h:0, d18:2^4t,8t^-C20h:0, d18:2^4t,8c^-C20h:0, t18:1^8c^-C22h:0, t18:1^8c^-C24h:0). Cells were stained by 1% CBB and photographed (20× magnification). Scale bar: 100 μm. (**B**) Neurite length distribution changes of PC12 cells incubated with 50 µM kCer or kCer molecular species (t18:1^8c^-C22h:0, t18:1^8c^-C24h:0, d18:2^4t,8t^-C16h:0, d18:2^4t,8c^-C16h:0, d18:2^4t,8t^ -C18h:0, d18:2^4t,8c^-C18h:0, d18:2^4t,8t^-C20h:0, d18:2^4t,8c^-C20h:0). All counting of the number of neurites with lengths of 10 to 150 μm is shown as 100% and calculated for each of x-axis valued neurite length.

**Figure 5 ijms-20-02116-f005:**
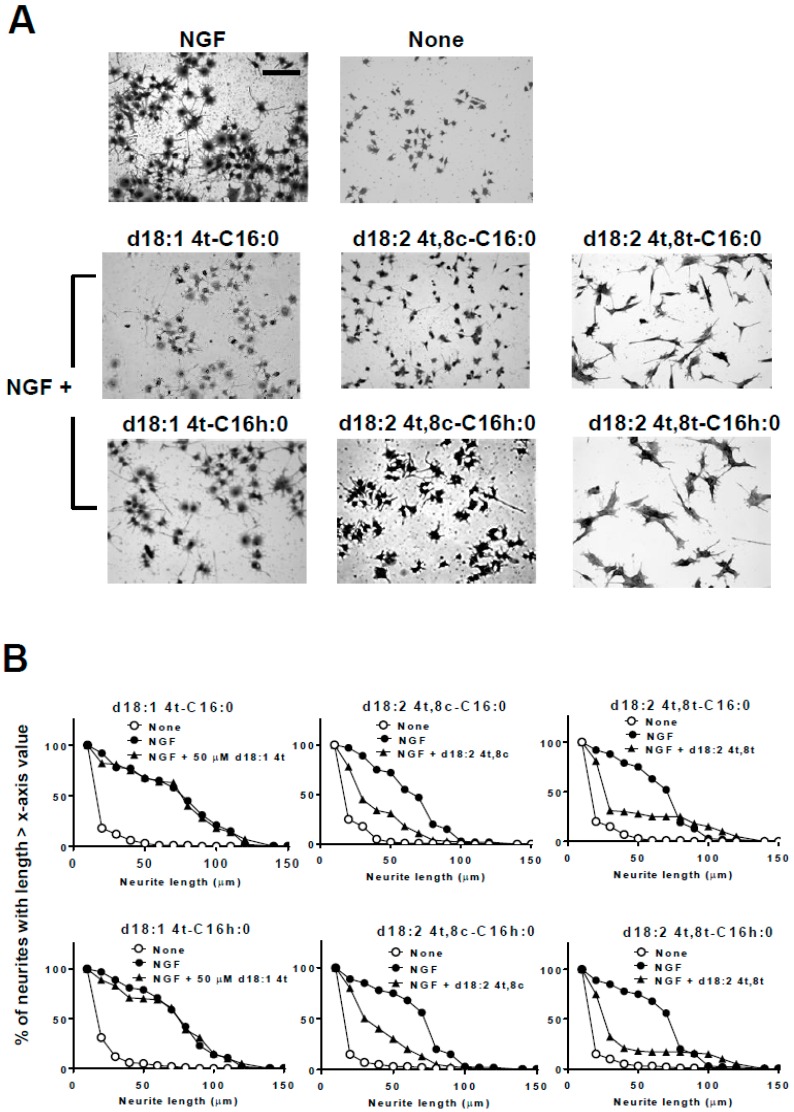
Effect of α-hydroxylation of C16 fatty acids and 8-trans or -cis unsaturation of sphingadienine on neurite outgrowth, neurite length distribution, and cell morphological changes in NGF-primed PC12 cells. (**A**) Representative images of NGF (100 ng/mL, 2.8 nM in medium)-primed PC12 cells (7.5 × 10^4^ per mL) incubated in the absence or presence of NGF and 50 µM kCer molecular species (d18:1^4t^-C16:0, d18:2^4t,8c^-C16:0, d18:2^4t,8t^-C16:0, d18:1^4t^-C16h:0, d18:2^4t,8c^-C16h:0, d18:2^4t,8t^-C16h:0 (). Cells were stained by 1% CBB and photographed (20× magnification). Scale bar: 100 μm. (**B**) Neurite length distribution changes of PC12 cells incubated with 50 µM kCer molecular species (d18:1^4t^-C16:0, d18:2^4t,8c^-C16:0, d18:2^4t,8t^-C16:0, d18:1^4t^-C16h:0, d18:2^4t,8c^-C16h:0, d18:2^4t,8t^-C16h:0). All counting of the number of neurites with lengths of 10 to 150 μm is shown as 100% and calculated for each of x-axis valued neurite length.

**Figure 6 ijms-20-02116-f006:**
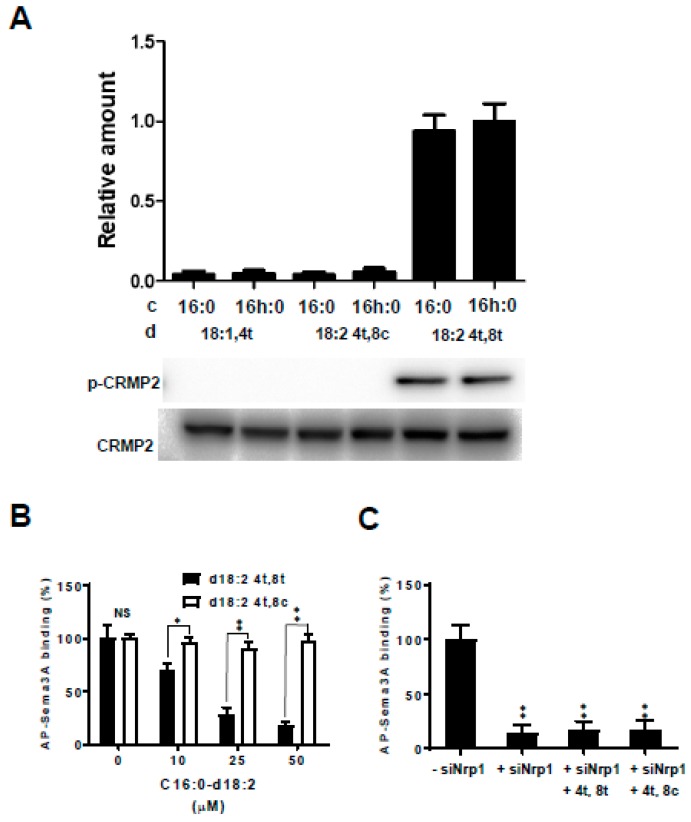
Effect of collapsin response mediator protein 2 (CRMP2) phosphorylation and binding activity of kCer molecular species with 8-trans unsaturated sphingadienine. (**A**) Comparison of CRMP2 phosphorylation in response to treatment with kCer molecular species (d18:1^4t^-C16:0, d18:1^4t^-C16h:0, d18:2^4t,8c^-C16:0, d18:2^4t,8c^-C16h:0, d18:2^4t,8t^-C16:0, d18:2^4t,8c^-C16h:0) in PC12 cells primed with NGF for 48 h and analyzed by western blotting. In the graph, c and d are fatty acid and sphingoid bases of the kCer molecular species, respectively. (**B**) Binding characteristics of Sema3A to the cell surface receptor in PC12 cells. Substitution of alkaline phosphatase-fused Sema3A (AP-Sema3A) bound to cell surface Nrp1 by kCer. Cells were cultured in 24-well microplates and treated with 15.0 APU of 25 nM AP-Sema3A in the presence of 10–50 µM kCer molecular species (d18:2^4t,8t^ -C16:0 or d18:2^4t,8c^-C16:0). Cells were washed and incubated at 65 °C for 30 min, and the remaining AP activity was measured using BCIP/NBT Phosphatase Substrate and ImageJ software. Data are shown as the mean ± SD (*n* = 4). * *p* < 0.05, ** *p* < 0.01 vs. d18:2^4t,8c^-C16:0 respectively, by an unpaired *t* test. NS, not significant. (**C**) Substitution effect of 50 µM d18:2^4t,8t^-C16:0 or d18:2^4t,8c^-C16:0 examined in untreated cells and siNrp1-treated cells. Data are shown as the mean ± SD (*n* = 4). ** *p* < 0.01 vs. control (−siNrp1) respectively by one-way ANOVA.

**Table 1 ijms-20-02116-t001:** Summary of chemical strucure /Sema3A-lkie activity relationship of kCer molecular species.

	Sema3A	kCer	kCer Molecular Species	Sphingoid Base	d18:1		d18:2 4t,8c				d18:2 4t,8t				t18:1 8c	
				Fatty Acid	C16:0	C16h:0	C16:0	C16h:0	C18h:0	C20h:0	C16:0	C16h:0	C18h:0	C20h:0	C22h:0	C24h:0
Neurite outgrowth inhibition	^1^ ++	++			-	-	^2^ +	+	+	+	++	++	++	++	^3^ -	-
Change of cell shape	^8^ spindle	spindle			^10^ NC	NC	^9^ round	round	round	round	spindle	spindle	spindle	spindle	NC	NC
pCRMP2	^4^ +	+			^5^ -	-	-	-			+	+			-	-
Substitution of cell receptor-Sema3A binding	^6,a^ +	^a^ +					^7^ -				+					

^1^ ++: presence of strong neurite outgrowth inhibitory activity, ^2^ +: presence of moderate neurite activity, ^3^ -: absence of neurite activity, ^4^ +: presence of CRMP2 phosphorylation activity, ^5^ -: absence of CRMP2 activity, ^6^ +: presence of binding activity, ^7^ -: absence of binding, ^8^ spindle, ^9^ round: cell morphological change after treatment, ^10^ NC: no change of cell shape, ^a^ kCer and Sema3A bindings are referred from [37].

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
