# Peer review of "Neurite Outgrowth and Morphological Changes Induced by 8-trans Unsaturation of Sphingadienine in kCer Molecular Species"

_ijms, 2019, doi:10.3390/ijms20092116_

Round 1
Reviewer 1 Report
Manuscript review:
"Neurite outgrowth and morphological changes induced by 8-trans unsaturation of sphingadienine in 2 kCer molecular species" by Usiki et al.
The topic of the manuscript addresses an interesting issue: the potential role of endogenous ceramide as supplement to improve the healthy life expectancy of the population. Recent studies have shown that food sphingolipid and especially from plant could have beneficial effect on several physiological parameters. This is the case of glucosylceramide (kGlCer) form the plant Konjac. It is know that plant sphingolipid contained different sphingoid bases than animal-types ceramide. Indeed, sphingoid base such as d18:2 and t18:1 are relatively abundant in plant but not in mammals. Up to date, the biological role of ceramide containing this specific sphingoid bases are not know. In the present study, the authors explore the role of various ceramide from Konjac (kCer). Interestingly, they found that kCer are biologic active and regulate neurite outhgrowth stimulated by NGF. It appeared that the nature of sphingoid base but not the presence of an a-hydroxylated fatty acid is important for their biological activities. Importantly, ceramide containing sphingosine (such as in mammals were inactive). This effect seemed to be mediated through the Sema3A receptor neurophilin 1 (Nrp1). Altogether, these results are interesting since they demonstrate for the first time that plant kCer in contrast to common ceramide in mammals could mediate specific biologic response and therefore could serve as potential therapeutic tools. Data presented are convincing even if the discussion section is weak for some part and a clear additional experiment will be crucial to support the role of Nrp1 in kCer-induced neurite outgrowth. See my comments below:
The authors found that kCer containing d18:2 4t,8t but also d18:2 4t,8c inhibited neurite extension induced by NGF. However, it appeared that both kCer have a different effect on the morphology of PC12 cells. More importantly, the authors found that kCer containing d18:2 4t,8t but not d18:2 4t,8c regulated phosphorylation of CRMP2, a factor essential for Sema3A activity.
Moreover, they found that kCer containing d18:2 4t,8t but not d18:2 4t,8c could inhibit binding of AP-SemaA3 to a cell surface receptor (probably Nrp1).
Based on these data, the authors concluded that kCer containing d18:2 4t,8t but also d18:2 4t,8c regulated neurite outgrowth but did not go deeper in their explanation. How explain the different effect of kCer containing d18:2 4t,8t and kCer containing d18:2 4t,8c kCer on PC12 morphology. This should be discussed more extensively in the discussion section.
The siNrp1 data are really interesting since they suggest that kCer containing d18:2 4t,8t mediated its effect through this receptor. However, the figure 6 is only an indirect evidence. In order to fully support a role of Nrp1 in the effect of kCer containing d18:2 4t,8t, the authors should determine the effect of kCer containing d18:2 4t,8t in PC12 cells where Nrp1 is down-regulated.
Minor comments:
- Supplemental figures are missing from the PDF file.
- Preparation of sphingoid base from kGlcCer is not described
- HaCat cells are not used in the present study
Author Response
Responses to Reviewers’ comments
We appreciate the reviewer’s comments. The responses to the reviewer #1 are described as
below. The revised sentences are shown in red-colored.
There was some high similarity index with Method sections in our previous publications. Therefore, we rephrased them (P12, L7-11 and L22-24; P13, and P14, L1-7; P15, L7-12).
Reviewer #1:
1.The authors found that kCer containing d18:2 4t,8t but also d18:2 4t,8c inhibited neurite extension induced by NGF. However, it appeared that both kCer have a different effect on the morphology of PC12 cells. More importantly, the authors found that kCer containing d18:2 4t,8t but not d18:2 4t,8c regulated phosphorylation of CRMP2, a factor essential for Sema3A activity.
>>kCer treatment causes disappearance of short neurites and remaining of long neurites. This change of neurite length-distribution reflects on the inhibition of neurite outgrowth by kCer treatment. Previously we have shown the relationship between neurite length and neurite outgrowth inhibition. We add this paper[37] to the discussion section (P9. L10).
2. Moreover, they found that kCer containing d18:2 4t,8t but not d18:2 4t,8c could inhibit binding of AP-SemaA3 to a cell surface receptor (probably Nrp1).
>>In our previous paper, we have already demonstrated the binding to Nrp1 by FITC-Sema3A. However, the relation to the sphingadienine is a new finding.
3. Based on these data, the authors concluded that kCer containing d18:2 4t,8t but also d18:2 4t,8c regulated neurite outgrowth but did not go deeper in their explanation. How explain the different effect of kCer containing d18:2 4t,8t and kCer containing d18:2 4t,8c kCer on PC12 morphology. This should be discussed more extensively in the discussion section.
>>We have shown the morphology is associated with microtubule depolymerization in our previous paper.The previous paper is added to the reference [37] in the Discussion section (P9, L10)..
4. The siNrp1 data are really interesting since they suggest that kCer containing d18:2 4t,8t mediated its effect through this receptor. However, the figure 6 is only an indirect evidence. In order to fully support a role of Nrp1 in the effect of kCer containing d18:2 4t,8t, the authors should determine the effect of kCer containing d18:2 4t,8t in PC12 cells where Nrp1 is down-regulated.
>>We agree with the reviewer’s comment. In our further study, we intend to test a direct binding of the active isomer containing d18:2 4t,8t of kCer species to Nrp1 by in vitro assay.
Minor comments:
- Supplemental figures are missing from the PDF file.
>>Sorry to be no attachment of supplement figures. We add the PDF files herewith.
- Preparation of sphingoid base from kGlcCer is not described
>>The preparation of sphingoid bases is shown in the supplemental figures.
- HaCat cells are not used in the present study
>>In the present study, HaCaT cells is not used. However, we mentioned it in the discussion, because it is important to understand the mechanism of Sema3A-like activity in PC12 cells, too.

Reviewer 2 Report
In the manuscript “Neurite outgrowth and morphological changes induced by 8-trans unsaturation of sphingadienine in kCer molecular species”, the authors prepared several plant-type ceramide species from konjac and examined these structure-activity relationships for inhibition of neurite outgrowth. They demonstrated that ceramides including 4t, 8t-sphingadienine show agonistic effects on neuropilin1 and inhibit NGF-induced neurite outgrowth. The experiments are well designed and the article is worth publishing in IJMS if the authors answer the following requests.
Major comments
1. Summarize the structure-activity relationships of the ceramide species tested for cell shape change and Sema3A-like activity including CRMP2 phosphorylation in a Table.
2. The text can be improved with English proofreading service. And the abstract should be further polished. Add the following explanation to the abstract.
a. Konjac ceramide is a plant-type ceramide species consisting of characteristic sphingoid bases.
b. Nrp1 (Neuropilin1) is a receptor for Sema3A.
c. kCer inhibits NGF-induced neurite phenotypes like Sema3A.
d. Delete 8-cis unsaturation of phytosphingosine (L27) because it is confused with 8-cis unsaturation of kCer molecular species that is more important.
Minor comments
1. Page3, L98: Add the following explanation at first that each kCer species was enzymatically synthesized from kGlcCer.
2. Page3, L123: Add the purpose to synthesize these sphingoid bases.
For example, add “To synthesize d18:24t,8t- and d18:24t,8c-C16:0 non-hydroxy ceramide,”
3. Page4, L133-4: Does “compared with” mean “like” or “unlike”?
“(Fig. 2B)” should be inserted after “molecular species”.
4. Page5, L158: Add the following phrase.
(d18:1 4t), which is a major sphingobase observed in mammals, had no effect
5. How does d18:24t,8c-ceramide affect neurite outgrowth inhibition although the species is not associated with Sema3A-like activity. Discuss it.
6. In Fig. 3B, 4B, and 5B, what does 100% indicate? Explain it. There is only a little difference between d18:24t,8t-ceramide and d18:24t,8c-ceramide in these graphs. Show or explain the differences more clearly in some way if possible.
7. Does dietary kCer circulate through the body and affect skin tissue at the effective concentration used in this study?
Author Response
Responses to Reviewers’ comments
We appreciate the reviewer’s comments. The responses to the reviewer #2 are described as
below. The revised sentences are shown in red-colored.
There was some high similarity index with Method sections in our previous publications. Therefore, we rephrased them (P12, L7-11 and L22-24; P13, and P14, L1-7; P15, L7-12).
Reviewer #2:
Major comments
1. Summarize the structure-activity relationships of the ceramide species tested for cell shape change and Sema3A-like activity including CRMP2 phosphorylation in a Table.
>> According to the reviewer’s comment, we add Table 1 in the Text (P17). We also add the following sentence, “Table 1 shows a summary of the structure -activity relationship of kCer molecular species. C16 and C18 fatty acid commonly carbon chain length enriched in plant ceramides” (P10, L25-27).
2. The text can be improved with English proofreading service. And the abstract should be further polished. Add the following explanation to the abstract.
>>We improve the text by English checking service after the revision.
a. Konjac ceramide is a plant-type ceramide species consisting of characteristic sphingoid bases.
>>According to the reviewer’s comment, we changed the sentence by the addition of the red-inked characters (in Abstract).
b. Nrp1 (Neuropilin1) is a receptor for Sema3A.
>> According to the reviewer’s comment, we changed the sentence by the addition of the red-inked characters (in Abstract).
c. kCer inhibits NGF-induced neurite phenotypes like Sema3A.
>>Sema3A inhibits NGF-induced neurite phenotype. kCer gives a similar phenotype to Sema3A. We do not understand the meaning of this comment.
d. Delete 8-cis unsaturation of phytosphingosine (L27) because it is confused with 8-cis unsaturation of kCer molecular species that is more important.
>> According to the reviewer’s comment, we have deleted it (in Abstract).
Minor comments
1. Page3, L98: Add the following explanation at first that each kCer species was enzymatically synthesized from kGlcCer.
>> We added the following sentence, “Each of kCer molecular species was enzymatically synthesized from kGlcCer.” (P6, L2).
2. Page3, L123: Add the purpose to synthesize these sphingoid bases.
For example, add “To synthesize d18:24t,8t- and d18:24t,8c-C16:0 non-hydroxy ceramide,”
>> We added the following sentence, ”Plant type-ceramides are mostly composed of a-hydroxyl fatty acid. To test whether the presence of a-hydroxylation of fatty acid is essential for the activity, we were forced to prepare plant-type sphingoid bases from plant ceramides, followed by synthesizing non-hydroxy plant ceramides using non-hydroxylated fatty acid” (P5, L16-19).
3. Page4, L133-4: Does “compared with” mean “like” or “unlike”? “(Fig. 2B)” should be inserted after “molecular species”.
>> “compared with” is changed to “like” (P6, L4). “(Fig. 2B)” is moved after “molecular species”(P6, L4).
4. Page5, L158: Add the following phrase.
(d18:1 4t), which is a major sphingobase observed in mammals, had no effect
>>We added the sentence according to the reviewer’s suggestion (P7, L2).
5. How does d18:24t,8c-ceramide affect neurite outgrowth inhibition although the species is not associated with Sema3A-like activity. Discuss it.
>> In the discussion (P9, L26-27 and P10, L1-2), we added the following sentence, “The inhibition of NGF-primed neurite outgrowth by C8 cis sphingadienine may be influenced by cell toxicity of this ceramide species. More than 50 mM, ceramides show some of cell toxicity by perturbating cell membrane. This influence is often induced by lipid structure-peculiar, non-specific manner due to the insolubility. “.
6. In Fig. 3B, 4B, and 5B, what does 100% indicate? Explain it. There is only a little difference between d18:24t,8t-ceramide and d18:24t,8c-ceramide in these graphs. Show or explain the differences more clearly in some way if possible.
>> We add a reference [37] into the Method section (P12, L15-16). We also insert the following sentence, “All counting number of neurites with lengths of 10 to 150 μm is shown as 100 % and calculated for each of x-axis valued-neurite length” after the sentence “…..cumulative ratio“ in the Method (L335). previously as described [37]”.
7. Does dietary kCer circulate through the body and affect skin tissue at the effective concentration used in this study?
>>There are a lot of undetermined mechanism after oral administration of plant ceramide-intestinal absorption, biotransformation, blood-barrier transporting, and distribution to skin tissue. We do not have data of kCer concentration in skin tissue. However, we have demonstrated transdermal administration of kCer to mice skin is effective.
